# Camrelizumab-based induction chemoimmunotherapy in locally advanced stage hypopharyngeal carcinoma: phase II clinical trial

Hongli Gong[1,3], Shu Tian[2,3], Hao Ding[2], Lei Tao[1], Li Wang[2], Jie Wang[2], Tian Wang[2], Xiaohui Yuan[1], Yu Heng[1], Ming Zhang[1], Yong Shi[1], Chengzhi Xu[1], Chunping Wu[1], Shengzi Wang[2] ✉ & Liang Zhou ✉[1] ✉

This phase II trial aimed to determine the efficacy and safety of induction chemoimmunotherapy of camrelizumab plus modified TPF in locally advanced hypopharyngeal squamous cell carcinoma (LA HSCC) (NCT04156698). The primary endpoint was objective response rate (ORR), and secondary endpoints were 3-year overall survival (OS), progression-free survival (PFS), larynx preservation rate (LPR), and metastasis-free survival (MFS). Patients (cT3-4aN0-2M0), regardless of sex, received induction chemoimmunotherapy for three cycles: camrelizumab 200 mg d1, docetaxel 75 mg/m² d1, cisplatin 25 mg/m² d1-3, and capecitabine 800 mg/m² bid d1-14, q21d. Patients were assigned to radioimmunotherapy if they had a complete or partial response, those with stable or progressive disease underwent surgery and adjuvant (chemo)radiotherapy. Camrelizumab was maintained post-radioimmunotherapy. Fifty-one patients were enrolled with a median follow-up duration of 23.7 months. After induction therapy, the ORR was 82.4% (42/51), meeting the prespecified endpoint. Grade 3/4 adverse events occurred in 26 patients, and no treatment-related death occurred. As three-year outcomes were immature, two-year OS, PFS and LPR were reported. As no distant metastatic event had occurred, MFS was not reported here. The two-year OS, PFS, and LPR rates were 83.0%, 77.1%, and 70.0%, respectively. The induction chemoimmunotherapy of camrelizumab plus TPF showed a high ORR rate with an acceptable safety profile in LA HSCC.

Hypopharyngeal carcinoma, an aggressive type of head and neck squamous cell carcinoma (HNSCC), arises from the mucosal epithelium of the pyriform sinus, postcricoid region, and posterior hypopharyngeal wall, with an estimated 84,254 new cases and an estimated 38,599 deaths in 2020 globally[1]. Owing to the hidden anatomical sites, this carcinoma is typically asymptomatic at an early stage, and

advanced disease is generally treated with surgery, radiation, and systemic therapy with curative intention[2].

Function preservation is one major problem that has to be considered when making a treatment decision. RTOG 91-11 trail is the landmark in clinical that establishes chemoradiotherapy for larynx preservation and locoregional control in patients with HNSCC[3].

[1]ENT institute and Department of Otorhinolaryngology, Eye & ENT Hospital, Fudan University, Shanghai 200031, China. [2]Department of Radiation Oncology, Eye & ENT Hospital, Fudan University, Shanghai 200031, China. [3]These authors contributed equally: Hongli Gong, Shu Tian. ✉e-mail: shengziwang@fudan.edu.cn; zhoulent@126.com

Induction chemotherapy of docetaxel, cisplatin, and fluorouracil, an evidence-based option for advanced HNSCC, generates a favorable response and milder toxicity profile and improves survival outcomes and function preservation from pooled data analyses (TAX 323, TAX 324, GORTEC 2000-01, and TTCC 2002)[4,5]. Recently, immune checkpoint inhibitors (ICIs), humanized anti-programmed death receptor 1 (PD-1) antibodies, have presented efficacy and safety in treating HNSCC. Patients with recurrent or metastatic (R/M) HNSCC after platinum chemotherapy treated with nivolumab monoclonal antibody have prolonged overall survival over standard and single-agent treatment[6]. Pembrolizumab monotherapy is well-tolerated and shows clinically substantial antitumor activity in patients with R/M HNSCC[7], with characteristics of prolonging overall survival and favorable safety profile compared with standard care therapy[8]. Camrelizumab (SHR-1210), another PD-1 monoclonal antibody, is a well-tolerated and appropriate option for patients with R/M nasopharyngeal carcinoma[9]. However, the objective response rate, survival outcome, laryngeal function preservation, and safety profile of these anti-PD-1 antibodies in locally advanced HNSCC patients remain uncertain.

The purpose of this trial was to determine the objective response rate, survival outcome, laryngeal function preservation, and safety profile of camrelizumab in combination with a modified TPF (docetaxel, cisplatin, and capecitabine) regimen of induction therapy for locally advanced hypopharyngeal squamous cell carcinoma (LA HSCC) patients who were previously untreated and required total laryngectomy.

## Results

### Patient characteristics

Between May 21, 2020, and November 15, 2023, 53 patients were screened, and 51 were enrolled and received induction chemoimmunotherapy with camrelizumab + TPF. Baseline demographics and clinical characteristics were recorded (Table 1 and Supplementary Table 1), and the study profile was summarized (Fig. 1). The median duration of follow-up from enrollment to the date of cutoff or death, whichever appeared first, was 23.7 months (range: 6.4–42.3 months).

### Radiological efficacy

Objective response rate (ORR) of these patients was evaluated after induction chemoimmunotherapy, with 42 (82.4%) patients exhibiting partial response (PR) and 9 (17.6%) patients exhibiting stable disease (SD) from the evaluation of Response Evaluation Criteria in Solid Tumors (RECIST) version 1.1 about primary target solid tumor shrinkage (Fig. 2A). Representative images of laryngoscopy and MRI of LA HSCC patients showing PR before and after induction chemoimmunotherapy were shown (Supplementary Fig. 1). Regarding SD population, eight patients (88.9%) were at T4a stage.

At the analysis time based on data cutoff, four patients remained on the treatment of camrelizumab maintenance. During the radiotherapy plus camrelizumab treatment, 31 patients completed, eight suspended camrelizumab treatment, and three suspended radioimmunotherapy. During the camrelizumab maintenance period, 15 patients completed treatment but there were 12 suspensions. Nine patients showed SD after induction chemoimmunotherapy; six underwent surgical treatment (five underwent total laryngectomy, and one underwent larynx-preserving surgery) with three pathological complete responses (CR), one experienced progress disease (PD), and two rejected surgery and requested radiotherapy. In patients who received induction therapy plus subsequent radioimmunotherapy, one patient developed a relapse of tuberculosis during radioimmunotherapy and subsequently suspended radiotherapy and received anti-tuberculosis treatment followed by total laryngectomy. Interestingly, the postoperative pathology of this patient was CR. None of the patients experienced distant metastasis. Of the 16 patients who lost laryngeal function, nine received total laryngectomy. Of the ten patients undergoing surgical treatment, six received surgery after induction chemoimmunotherapy, two underwent salvage total laryngectomy during radiotherapy, and two underwent salvage total laryngectomy owing to uncontrolled disease during camrelizumab maintenance (Supplementary Data 1). Eight patients died at the time of analyses: five died from disease progression, two from uncontrolled disease, and one died from local recurrence and regional metastasis (Fig. 2B).

### Safety profiles

Treatment-related Adverse events (AEs) were recorded, leading to treatment suspension in 11 (21.6%) of 51 patients. The most common AE was alopecia (100%), followed by reactive cutaneous capillary endothelial proliferation (RCCEP) (90.2%) (Supplementary Fig. 2), nausea/vomiting (80.4%), and fatigue (58.8%) in the period of induction therapy of camrelizumab + TPF. Twenty-six (51.0%) of 51 patients experienced grade 3 or 4 induction therapy-related AEs, including RCCEP, pneumonia, rash, leukopenia, neutropenia, thrombocytopenia, anemia, myalgia, diarrhea, increased creatine kinase, and tuberculosis recurrence. The AEs during the treatment period were also recorded (Table 2 and Supplementary Table 2). None of the AEs led to death.

### Secondary endpoints

Of note, the preplanned timeframe for secondary endpoints was 3 years, however, these outcomes are immature as the median duration of follow-up was only 23.7 months. Therefore, 1- and 2-year outcomes are reported. This change was approved by our Data Safety Monitoring Board. As no distant metastasis occurred at the time of analysis, MFS was not reported here. The estimated 1-year overall survival (OS) rate, progression-free survival (PFS) rate, and larynx preservation rate (LPR) were 93.7% (95% confidence interval [CI], 81.7–97.9%), 85.3% (95% CI, 71.6–92.7%), and 75.0% (95% CI, 60.2–85.0%), respectively. The estimated 2-year OS, PFS, and LPR rates were 83.0% (95% CI, 67.4–91.6%), 77.1% (95% CI, 61.4–87.1%), and 70.0% (95% CI, 51.0–78.8%), respectively (Fig. 3). After induction chemoimmunotherapy, PR patients underwent radioimmunotherapy, and SD patients underwent surgery plus adjuvant therapy. The estimated 1-year OS and PFS rates in patients with PR were 86.9% (95% CI, 68.7–94.9%) and 79.8% (95% CI, 61.9–89.9%). The estimated 2-year OS and PFS rates in patients with PR were 80.0% (95% CI, 55.9–91.7%) and 72.8% (95% CI, 50.1–86.5%). Only nine patients showed SD, and the estimated 1-year and 2-year OS and PFS rates in these patients were all 64.7% (95% CI, 25.6–87.0%) (Supplementary Fig. 3). No significant difference was observed in survival rates between these two groups ($p = 0.055$ and $p = 0.185$). LPR rate in T3 stage patients was higher in patients with T4 stage ($p = 0.040$). LPR rate in patients without esophagus involved was higher than that of patients with esophagus involved ($p = 0.006$) (Fig. 3). Another exploratory subset analysis determined that no significant differences in OS, PFS, and LPR were observed in subgroups stratified by T stage, N stage, esophagus involved, ENE or cartilage involvement.

### Exploratory analyses

Regarding PD-L1 expression analyses, 21 patients had adequate tumor tissue samples, and the CPS was evaluated (Fig. 4A). The primary tumor response levels in patients with CPS < 1 were not different from those with CPS ≥ 1 ($p = 0.211$). Further, ORR according to PD-L1 expression was evaluated, and there was no statistically significant association between ORR and CPS ($p = 0.257$) (Table 3). The OS and PFS rates according to PD-L1 expression were assessed, and no statistically significant difference was detected between these two subgroups ($p = 0.349$ and $p = 0.759$) (Supplementary Fig. 3).

Baseline tumor tissues from 23 patients were eligible for WES analyses. The most frequently mutated gene was *TP53* (20 of 21, 87%),

**Table 1 | Baseline clinical characteristics of 51 patients with LA HSCC**

|  | *N* = 51 | % |
|---|---|---|
| **Age (years)** | | |
| Median | 57.0 | |
| Range | 35–69 | |
| **Sex** | | |
| Male | 51 | 100 |
| Female | 0 | 0 |
| **Tumor site** | | |
| Pyriform sinus | 39 | 76.5 |
| Postcricoid | 3 | 5.9 |
| Posterior hypopharyngeal wall | 9 | 17.6 |
| **T stages 7th** | | |
| T3 | 26 | 51.0 |
| T4 | 25 | 49.0 |
| **N stages 7th** | | |
| N0 | 1 | 2.0 |
| N1 | 14 | 27.5 |
| N2b | 20 | 39.2 |
| N2c | 16 | 31.4 |
| **ENE** | | |
| ENE positive | 15 | 29.4 |
| ENE negative | 36 | 70.6 |
| **Esophagus** | | |
| Involved | 17 | 33.3 |
| No involved | 34 | 66.7 |
| **Thyroid cartilage** | | |
| Involved | 17 | 33.3 |
| No involved | 34 | 66.7 |
| **Vocal cord mobility** | | |
| Mobility | 23 | 45.1 |
| Impaired | 9 | 17.6 |
| Fixation | 19 | 37.3 |
| **CPS** | | |
| <1 | 13 | 25.5 |
| >1 | 8 | 15.7 |
| Not available | 30 | 58.8 |
| **ECOG** | | |
| 0 | 32 | 62.7 |
| 1 | 19 | 37.3 |

Source data are provided as a Source Data file.
*ENE* extranodal extension, *CPS* combined positive score, *ECOG* Eastern Cooperative Oncology Group.

followed by *MCU16* and *PIK3CA* (17 of 23, 74%). No significant differences in gene mutations, classic pathway enrichment, or tumor mutational burden (TMB) were observed between patients with PR and patients with SD (Fig. 4C). As eight of nine patients who experienced SD were at the T4 stage, we assessed the TMB in the T4 stage population. We found that patients with PR showed higher Muts/Mb than those of patients with SD (Fig. 4D). Regarding circulating lymphocyte cells, we found that the percentage of CD8+ in CD3 + CD19- total T lymphocytes was higher in PR patients than those of SD patients at T4 stage (Fig. 4G). Additionally, peripheral circulating lymphocyte analyses between PR and SD patients were also analyzed before chemoimmunotherapy and no difference was found (Supplementary Fig. 4). Further, the percentage of CD123 + CD303+ plasmacytoid

dendritic cells in total CD11c- cells in patients with SD was higher than those of PR patients at T4 stage (Fig. 4I, J).

## Discussion

ICIs targeting the PD-1/PD-L1 pathway augment endogenous antitumor responses, and several ICI trials are conducted to treat the R/M HNSCC and find promising outcomes. Radiotherapy may improve responses to immune therapy by shaping the tumor microenvironment and impacting the affluence and composition of tumor-infiltrating immune cells[10]. However, the KEYNOTE-412 and PembroRad trials revealed that patients with LA HNSCC did not benefit more from pembrolizumab plus chemo/radiotherapy than chemo/radiotherapy alone in event-free survival and locoregional control, and the JAVELIN Head and Neck 100 reported that avelumab plus chemoradiotherapy did not prolong median progression-free survival for LA HNSCC patients compared with chemoradiotherapy (14·6 months vs. 14·8 months)[11–13]. Besides, nivolumab alone or plus stereotactic body radiation therapy (SBRT) in R/M HNSCC patients yields similar ORR rates with 34.5% and 29.0%[14]. The ORR rate in nivolumab-treated patients with R/M HNSCC is 13.3% in the CheckMate 141 study[6]. The mean major pathological response (MPR) in neoadjuvant immunotherapy was 9.7%, and the pathological CR rate is 2.9%[15] from a meta-analysis, lower than recently reported data. Induction therapy (camrelizumab plus TP) followed by surgery shows promising oncologic outcomes in 30 patients with LA HNSCC, with an ORR of 96.7%, a pathological CR of 37.0%, and an MPR of 74.1%. It is similar to an induction therapy study of a combination of durvalumab with hypofractionated SBRT[16,17]. Camrelizumab and apatinib, followed by surgery in 20 patients with oral LA SCC, resulted in an MPR of 40%[18]. Pembrolizumab prior to surgery generated a pathologic response of 39% and an MPR of 7% in 92 patients with LA HNSCC[19]. A recent study analyzed a group of 27 patients treated with paclitaxel, cisplatin, and toripalimab and generated ORR of 85.2% and 1-year OS, PFS and LPR with 84.7%, 77.6% and 88.7%, respectively[20]. In this study, induction therapy with camrelizumab plus TPF in patients with LA HSCC, the ORR was 82.4%. Interestingly, ten patients underwent surgical treatment: six due to primary tumor exhibiting SD, three due to tumor uncontrolled, and one due to a relapse of tuberculosis during radiotherapy. Four (40%) patients displayed pathological negative results postoperatively with CR. The SD evaluation was assessed using an MRI scan and calculated using the RECIST system. This controversial estimation may be due to the MRI system misjudging tumor load, inflammation, mucosal edema, and fibrosis after immunotherapy, and further evaluation system is required to resolve this problem urgently.

Approximately 60% of patients with HNSCC present with advanced stage at initial diagnosis; these patients carry a high-risk of locoregional recurrence and distant metastasis and experience poor survival outcomes after definitive therapy[2]. Chemoradiotherapy chosen for advanced HNSCC patients with uninvolved cartilage imagination has good efficacy and quality of life, while salvage laryngectomy is preferred for patients with disease progression and relapse with severely impaired function. Five-year follow-up of the TAX 324 data generated long-term oncologic benefits in induction chemotherapy with TPF than PF, with higher 5-year OS (52% versus 42%, $p = 0.014$) and better 5-year PFS (45% versus 34%, $p = 0.011$)[21]. Regarding hypopharyngeal and laryngeal carcinoma, a longer median duration PFS (20.9 versus 10.1 months, $p = 0.037$) was determined in TPF over PF in induction chemotherapy[21]. Concerning organ preservation, a 10-year follow-up of the GORTEC 2000-01 study observed that 5- and 10-year larynx preservations were 74.0% vs. 58.1% and 70.3% vs 46.5% ($p = 0.01$) in induction chemotherapy with TPF over PF, respectively[22]. Induction therapy of modified TPF displayed promising efficacy compared with TP and TPF for LA HSCC (ORR: 64.5%, 45.1%, and 55.2%, respectively; 3-year LPR: 51.2%, 36.6%, and 31.8%, respectively, $p = 0.03$) in a previous study[23]. Cisplatin-based chemoradiotherapy is effective when patients

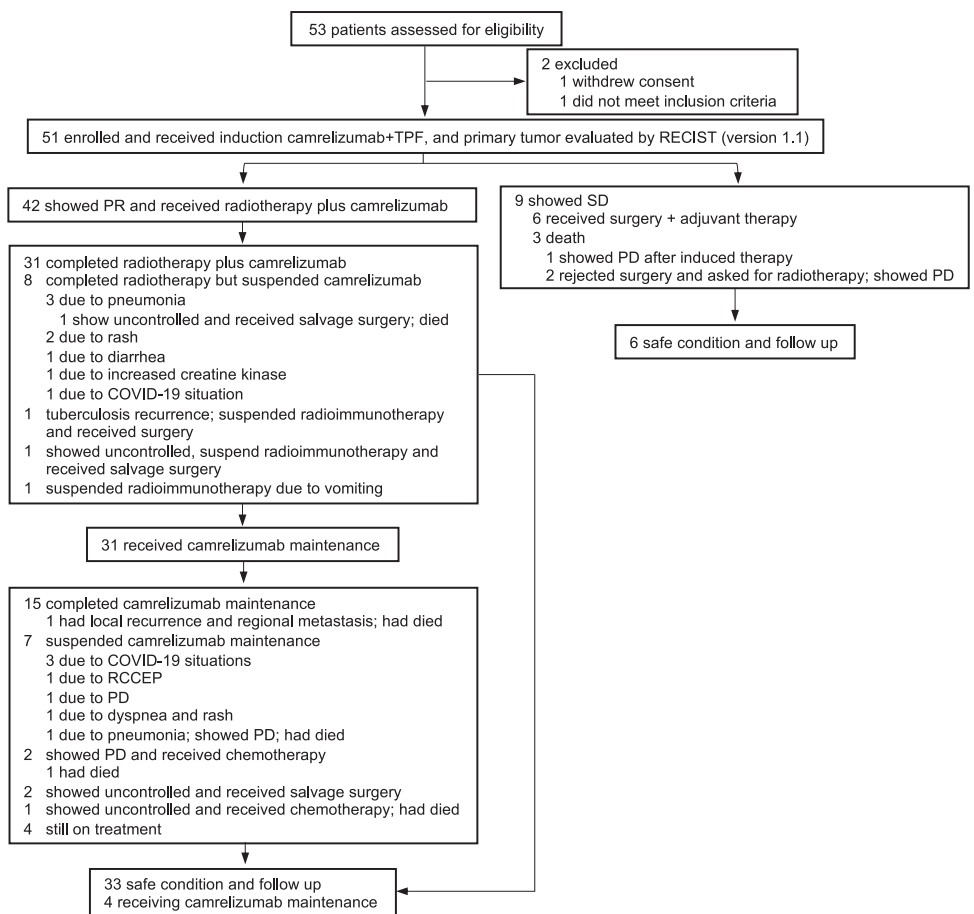

**Fig. 1 | Trial profiles of the 53 patients.** RECIST response evaluation criteria in solid tumors, RCCEP reactive cutaneous capillary endothelial proliferation, TPF docetaxel, cisplatin, and capecitabine, CR complete response, PR partial response, SD stable disease, PD progress disease. Source data are provided as a Source Data file.

get CR or PR after induction therapy. This study was designed in 2019 when several trials were initially designed to evaluate the antitumor activity and safety of ICIs plus radiotherapy/chemoradiotherapy for LA HNSCC[11–13]. We planned to assess the efficacy and safety of ICIs plus radiotherapy for CR/PR patients and surgery plus postoperative chemoradiotherapy for SD/PD patients after induction therapy. The OS and PFS rates of patients undergoing radioimmunotherapy may be better than patients undergoing surgery plus adjuvant therapy after induction therapy, but no statistical difference was detected. The discussion of the best practices for HNSCC remains ongoing, and this study found 2-year OS, PFS, and LPR rates of 83.0%, 77.1%, and 70.0%, after definitive therapy, respectively; these oncologic outcomes may be comparable to previous studies after long-term follow-up.

By immunohistochemistry evaluation, PD-L1 expression degrees in the HNSCC tumor microenvironment may be a screening method for chosen immune treatment and predict oncologic outcomes. The KEYNOTE-012, 040, and 048 studies determined a meaningful prolongation of survival treated by pembrolizumab in R/M HNSCC patients with more than 1% PD-L1 expression[7,8]. Besides, the KEYNOTE-040 study revealed that the median OS was 11.6 months with pembrolizumab, better than standard care with 6.6 months in patients with a PD-L1 tumor proportion score of 50% or higher[8]. The KEYNOTE-048 study detected that pembrolizumab prolonged OS compared with cetuximab with PF in the CPS of 20 or more patients with R/M HNSCC (median 14.9 vs. 10.7 months, $p = 0.0007$)[24]. Four-year follow-up results were recently updated and further demonstrated that patients with R/M HNSCC who received pembrolizumab treatment had superior OS in the CPS ≥ 20 and CPS ≥ 1 population[25]. However, the Check-Mate 141 found a negative association between patients with a tumor

PD-L1 expression level and the magnitude of effect of nivolumab therapy. Another study indicated that pathologic PD-L1 expression alone may not directly predict long-term survival outcomes[19]. In this study, the survival outcomes of different statuses of CPS in patients with LA HSCC were similar, indicating the efficacy of PD-1 inhibitors plus chemotherapy in induction therapy irrespective of PD-L1 expression results.

The PD-1 antibodies have been demonstrated to be safe and well-tolerated in previous studies. Pembrolizumab and nivolumab treatment-related any grade of AEs occur from 589–63%, and grade three or worse events occur ranging from 13–55%[6–8]. The most common events of any grade are fatigue, anemia, constipation, pruritus, rash, and hypothyroidism, while the most common events of grade three or worse are anemia, fatigue, alanine aminotransferase increase, aspartate aminotransferase increase, hyponatremia, and hypothyroidism[3,7,24,25]. Four, 25, and two patients died from AEs in the KEYNOTE-040, KEYNOTE-048, and Checkmate 141, respectively, but no treatment-related deaths occurred in the KEYNOTE-012[7,8,24]. Besides, pembrolizumab plus chemotherapy-related any grade of AEs was 75%, and grade three or worse events was 47%, which were higher than those of monoclonal antibody treatments[24]. During induction treatment of TP plus toripalimab, 25.9% of patients showed grade three or worse AEs, and the most common AE was increased aspartate aminotransferase (11.1%)[20]. A previous study showed that induction therapy of modified TPF displayed acceptable safety compared with TP and TPF; the ratio of grade 3/4 hematotoxicity was 19.3%, 7.8%, and 27.6%, respectively, and the ratio of grade 3/4 diarrhea was 1.6%, 0%, and 3.4%, respectively[23]. In this study, 51% of patients experienced grade 3/4 induction therapy-related AEs; the incidence of RCCEP was 90% and

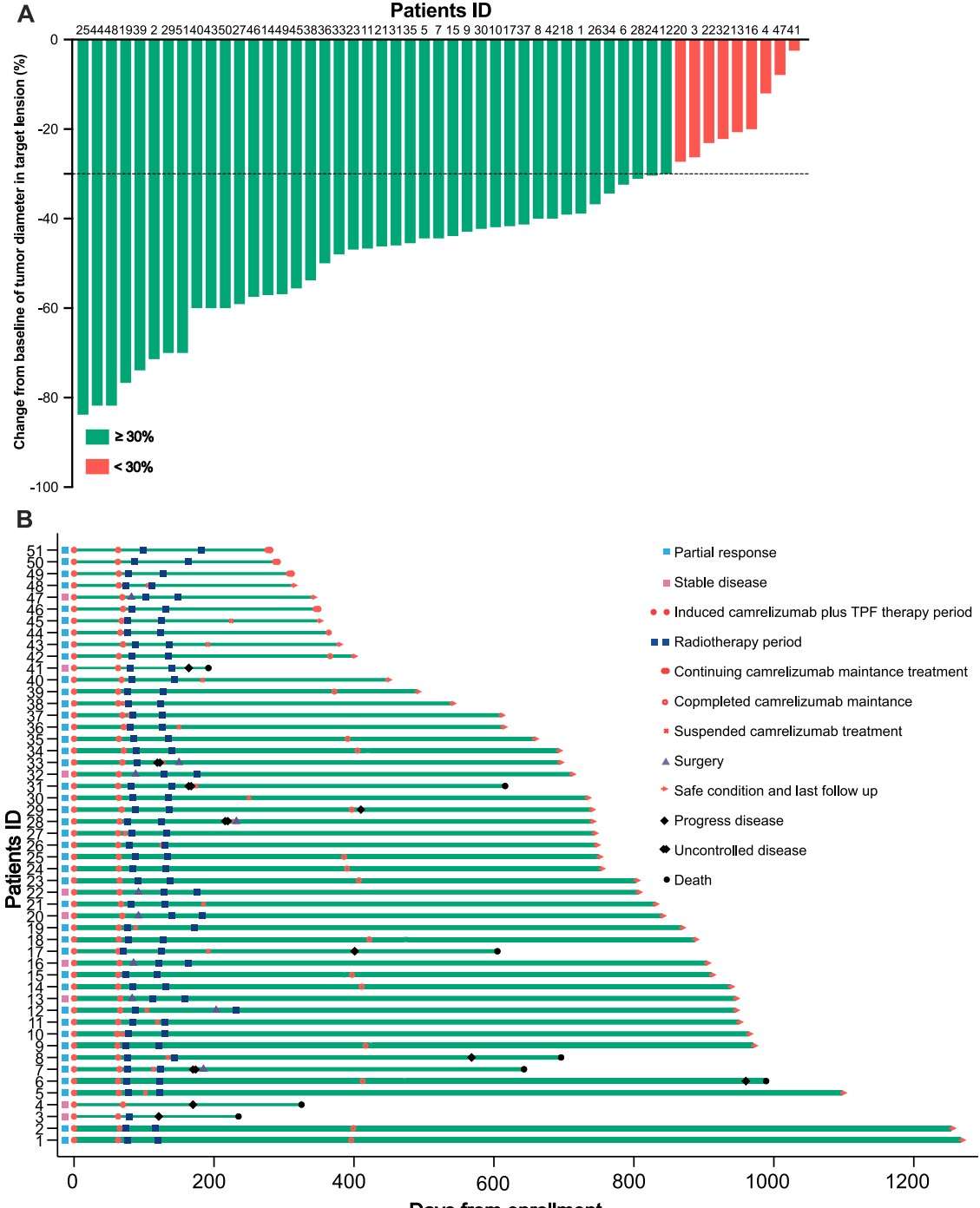

**Fig. 2 | Tumor response and treatment exposure in 51 LA HSCC patients.**
**A** Percentage reduction of tumor diameter of the primary lesion compared with baseline with Response Evaluation Criteria in Solid Tumors (RECIST) version 1.1 measurement. **B** Treatment exposure and response duration of induction chemoimmunotherapy, radioimmunotherapy, and immunotherapy maintenance in 51 patients. Source data are provided as a Source Data file.

tended to be higher than those of the previous studies, ranging from 58–88%[9,26]. The potential pathway of camrelizumab promoting RCCEP is activating CD4 + T cells, increasing IL-4 levels in T helper 2 cytokines, and stimulating CD163 + M2 macrophage differentiation and capillary endothelial cell proliferation by releasing vascular endothelial growth factor[27]. Interestingly, the event of RCCEP may generate a promising objective response rate and better survival outcomes, and this occurrence may be used as a factor in predicting the efficacy of camrelizumab immunotherapy for HNSCC patients[27,28]. These events were curable, and no drug AE-related deaths occurred in our study. Our

observation is consistent with previous studies, supporting that camrelizumab plus TPF possesses a favorable safety profile in clinical treatment.

*TP53* mutation is likely to correlate with high TMB, and these factors may predict therapeutic response and prognosis for patients with HNSCC[29]. R/M HNSCC patients with high TMB, determined at cutoffs of TMB ≥ 10 mut/Mb or TMB ≥ 12 mut/Mb, experience promising survival compared with patients with low level[30]. However, ref. 31 detected that a low TMB was correlated with better OS. Still, no correlation was observed between TMB and ORR or PFS in patients

**Table 2 | Induction chemoimmunotherapy-related adverse events in the intention-to-treat population**

| Adverse events | Grade 1–2 | Grade 3 | Grade 4 |
|---|---|---|---|
| RCCEP | 45 (88.2%) | 1 (2.0%) | 0 |
| Alopecia | 51(100%) | 0 | 0 |
| Pneumonia | 1(2.0%) | 1(2.0%) | 0 |
| Fever | 4 (7.8%) | 0 | 0 |
| Rash | 6 (11.8%) | 1 (2.0%) | 0 |
| Fatigue | 30 (58.8%) | 0 | 0 |
| Nausea/vomiting | 41 (80.4%) | 0 | 0 |
| Hypothyroidism | 3 (5.9%) | 0 | 0 |
| Thyroid stimulating hormone concentration decrease | 6 (11.8%) | 0 | 0 |
| Leukopenia | 11 (21.6%) | 4 (7.8%) | 0 |
| Neutropenia | 6 (11.8%) | 7 (13.7%) | 0 |
| Thrombocytopenia | 3 (5.9%) | 3 (5.9%) | 0 |
| Anemia | 20 (39.2%) | 3 (5.9%) | 0 |
| Increased ALT | 2 (3.9%) | 0 | 0 |
| Increased AST | 1(2.0%) | 0 | 0 |
| Total bilirubin elevation | 3 (5.9%) | 0 | 0 |
| Conjugated bilirubin concentration elevation | 2 (3.9%) | 0 | 0 |
| Albuminuria | 6 (11.8%) | 0 | 0 |
| Increased creatine kinase | 0 | 0 | 1(2.0%) |
| Increased urea nitrogen | 11 (21.6%) | 0 | 0 |
| Arthralgia/Myalgia | 0 | 1 (2.0%) | 0 |
| Constipation | 3 (5.9%) | 0 | 0 |
| Diarrhea | 8 (15.7%) | 3 (5.9%) | 0 |
| Tuberculosis recurrence | 0 | 1 (2.0%) | 0 |
| Hand-foot syndrome | 2 (3.9%) | 0 | 0 |
| Hypokalemia | 2 (3.9%) | 0 | 0 |

Source data are provided as a Source Data file.
*RCCEP* reactive cutaneous capillary endothelial proliferation, *ALT* alanine aminotransferase, *AST* aspartate aminotransferase.

with RM HNSCC treated with pembrolizumab and cabozantinib. Another study deciphered no difference in TMB between MPR and non-MPR patients with LA HNSCC treated by induction camrelizumab and apatinib[18]. In this study, we determined the TMB in 23 patients with sufficient tumor tissues available for WES and found that *TP53* was the most frequently mutated gene. Due to SD patients being at the T4 stage in this subgroup, we further evaluated TMB at the T4 stage and detected that PR patients had a higher TMB level than SD patients. Furthermore, we found that these PR patients had a high percentage of CD8+ T cells and a low percentage of CD123 + CD303+ in total CD11c-cells. DCs can induce a CD8+ T cell response with antitumor activity, leading to their activation and differentiation and causing tumor regression. In contrast, low level of CD8+ T cells may have feedback to stimulate activation of DCs[32]. However, this observation from limited clinical tumors needs to be confirmed by further study. There is an ongoing requirement for reforming strategies to select patients for chemoimmunotherapy, and high TMB and CD8+ T cell may be predictive biomarkers for pre-treatment efficacy.

This trial has a few limitations that have to be considered. It was a single-center, single-arm, and non-randomized clinical study. Only males were enrolled as hypopharyngeal cancer predominantly affects male population. No analysis based on sex or gender was conducted. The efficacy of camrelizumab pus modified TPF compared with a regimen of TPF in a multicenter and randomized control study is unknown. The long-term oncologic outcomes of this regimen remain

uncertain because the last patient was entered earlier this year. Nevertheless, this trial is ongoing, and the future information will be followed up and updated.

In sum, this study conducted the induction therapy of camrelizumab plus TPF in 51 patients with LA HSCC and generated a favorable ORR rate. After definitive radioimmunotherapy and maintenance, the OS, PFS, and LPR rates were also promising. These results suggest that the regimen of camrelizumab plus TPF is an effective treatment option with good tolerance, bringing novel survival outcomes and curable AEs.

## Methods

### Study design and participants
This open-label, single-arm, phase II, prospective, and single-center study was performed at Eye & ENT Hospital, Fudan University. The aim of the study was to assess the antitumor activity and safety profile of camrelizumab plus TPF of induction chemoimmunotherapy for newly diagnosed LA HSCC patients. Enrolled patients included the TNM stage of cT3-4aN0-2M0 (AJCC 7th edition).

Patients were enrolled if they met these criteria: with the curable purpose of local therapies as total laryngectomy; aged 18–70 years; without distant metastasis; no prior anti-cancer therapy; an Eastern Cooperative Oncology Group (ECOG) performance status of 0–2; an expected survival of more than 6 months; and adequate marrow and organ function. The critical exclusion criteria were bi-primary carcinoma, active autoimmune disease, immune deficiency, hepatitis B virus (HBV) or hepatitis C virus (HCV) infection, and active tuberculosis (TB). Of note, patients were collected regardless of sex or gender during the enrolled period, and there was no sex- and gender-based analyses in this study.

This trial was performed in accordance with the Declaration of Helsinki and the International Conference on Good Clinical Practice guidelines. The Ethics Committees of Eye & ENT Hospital, Fudan University, approved the trial protocol and treatments. All patients provided written informed consent before enrollment. This trial was registered with ClinicalTrials.gov, No. NCT04156698. The first patient was enrolled on May 25, 2020; cohort enrollment was completed on January 16, 2023. The interim analysis date of this study was April 15, 2023, and the last follow-up date for present study was November 15, 2023. The Committee of the Data and Safety Monitoring Board at Eye & ENT Hospital authorized the interim analysis, and authorization was obtained. The treatment and follow-up of this trial is ongoing.

### Procedures
Enrolled patients received induction camrelizumab + TPF every 3 weeks for three cycles. Camrelizumab was given 200 mg on day one intravenously. Participants also received docetaxel 75 mg/m$^2$ on day one intravenously, cisplatin 25 mg/m$^2$ on days 1–3 intravenously, and capecitabine 800 mg/m$^2$ twice daily on days 1–14 oral use. Patients were assigned to radical radiotherapy plus concurrent immunotherapy if they had a CR or PR or transferred to surgery followed by adjuvant (chemo)radiotherapy if they had SD or PD. The response was defined using RECIST (version 1.1). Radiotherapy was performed using Intensity-modulated Radiation Therapy. The dose of the primary site: Gross Tumor Volume (GTV) dose 66 (2.2 Gy/fraction)–70 Gy (2 Gy/fraction); Clinical Target Volume (CTV) 1.6–1.9 Gy/fraction. Radiotherapy of the cervical lymph node was the same as the primary lesion. Concurrent camrelizumab immunotherapy was given 200 mg intravenously every 3 weeks for three cycles. After completing radioimmunotherapy, camrelizumab maintenance was administered 200 mg every 3 weeks up to 18 cycles. When further evidence of progression occurred, camrelizumab was discontinued. Treatment could be discontinued at the patient's or oncologist's discretion. Surgery was conducted if patients had uncontrolled disease or locoregional recurrence. Patients with high-risk factors after total laryngectomy

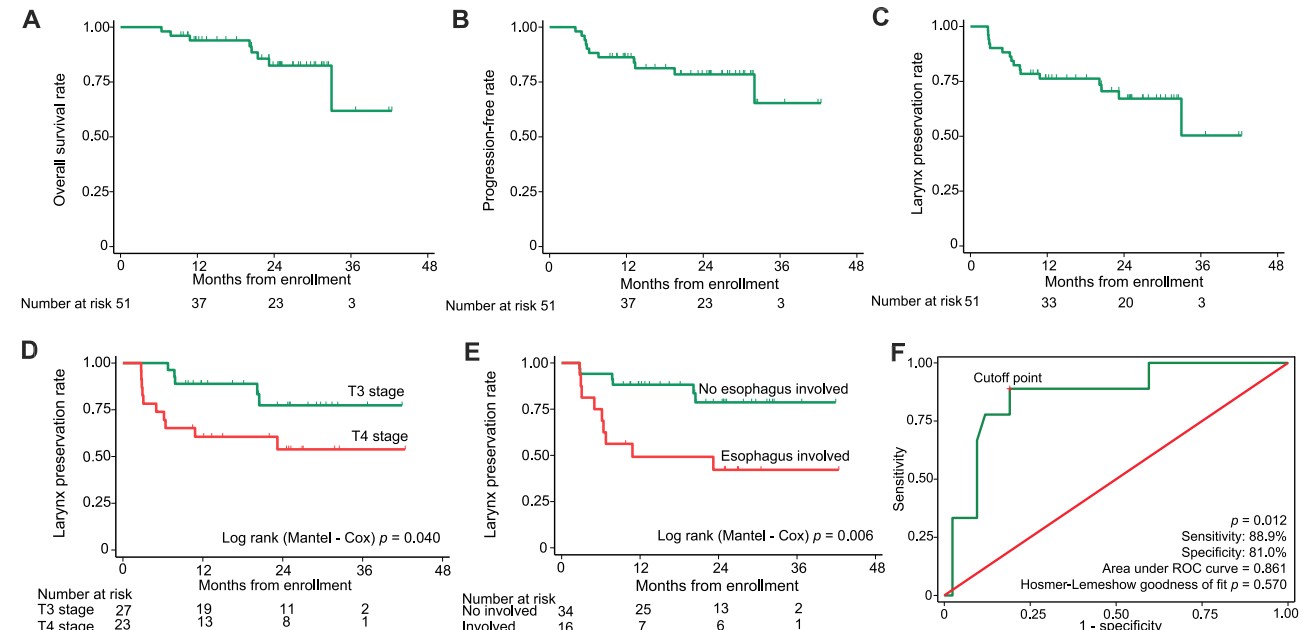

**Fig. 3 | Survival curves and objective response rate (ORR) predicting model.** Kaplan-Meier estimates of overall survival (**A**), progression-free survival (**B**), and larynx preservation rate (**C**) of 51 patients with LA HSCC. Larynx preservation rate of 50 patients with T3 stage and T4 stage (**D**) and patients with esophagus involvement and without esophagus involvement (**E**). **F** Multivariable model predicting ORR of induction chemoimmunotherapy of patients with LA HSCC based on pre-treatment tumor diameter (cutoff value with 37.5 mm; OR = 10.2, 95%CI [1.2–100.5]), T stage (OR = 14.4, 95% [1.6–126.4]) and esophagus involvement (OR = 5.6, 95%CI [1.2–26.5]) (*n* = 51 cases), and receiver operating characteristic curve illustrating the discrimination ability of this model. Source data are provided as a Source Data file.

were treated with adjuvant chemoradiotherapy using cisplatin (25 mg/m² on days 1–3, every 3 weeks).

AEs and laboratory safety were monitored before each treatment initiation and graded using the National Cancer Institute Common Terminology Criteria for Adverse Events (version 5.0). When patients experienced grade 3–4 drug-related AEs, camrelizumab treatment was withheld until toxicity resolved to grade 0–1. When drug-related toxicity had not resolved within 3 months of the last camrelizumab administration, patients had to discontinue the study treatment.

To evaluate response, contrast-enhanced MRI or CT scans were conducted at baseline and 2 month intervals after therapy initiation. Patients were followed up every 3 months.

### Outcomes
The primary endpoint was the ORR after three cycles of induction chemoimmunotherapy, defined as the proportion of patients with the best response, complete or partial response, as in the RECIST 1.1 evaluation. Secondary efficacy endpoints were 3-year OS, defined as the time from initial treatment to death due to any cause; 3-year PFS, defined as the time from initial treatment to the first documented disease progression or death due to any cause, whichever occurred first; 3-year LPR, defined as the time from initial treatment to a total laryngectomy, dysfunction of the larynx, or death; 3-year MFS, defined as the time from initial treatment to distant metastasis or death due to any cause. Of note, the planned timeframe for secondary endpoints was 3 years. As the median duration of follow-up was only 23.7 months. 1- and 2-year outcomes are reported here and this change was approved by our Data Safety Monitoring Board. Additional prespecified endpoints involved safety profiles, including clinical examination and AE, and investigators evaluated whether an AE was potentially immune-related. Exploratory analyses of whole-exome sequencing (WES) and peripheral blood analyses were performed.

### WES and whole peripheral blood analyses
Formalin-fixed paraffin-embedded and matched blood samples were collected from 23 patients who underwent biopsy at the Eye & ENT Hospital; these patients had sufficient tissue of tumor samples. Tissue DNA was extracted with GeneRead DNA FFPE Kit (180134, QIAGEN), and blood DNA was extracted with Magnetic Universal genomic DNA Kit (DP705, TIANGEN). The library construction and capture experiments used an Agilent kit and SureSelect V6 capture probes. Illumina NovaSeq PE150 sequencing was performed after the library inspection qualified. Somatic SNV INDEL calling was performed to manage the exome variant. The main logic of the tumor mutational burden (TMB) calculation was from the Focr research project. The CCDs' length was 32.4 M bases, and the types of mutations involved in the analyses include missense mutations, nonsense mutations, in-frame deletions, in-frame insertions, frameshift deletions, and frameshift insertions. The variant allele frequency was greater than or equal to 0.05, the reads support mutation was greater than or equal to 3, and the depth of the tumor was greater than or equal to 25.

Whole peripheral blood samples were centrifuged at 1962 g (3000 rpm) for 10 min to remove plasma, and the remaining blood cells were lysed with RBC lysis buffer (4300, eBioscience) at room temperature for 5 min. Cells suspension was washed in staining buffer and stained with fluorochrome-conjugated monoclonal antibodies for cell surface markers to identify different lymphocytes and their subgroups. Zombie UV Fixable Viability kit (423107, BioLegend) was used to assess live status. Cells were first incubated with the live/dead dye for 20 min at room temperature, then were washed and stained for 30 min at room temperature in the dark for cell surface markers to attach. Antibodies including PD-L1 (DAKO Cat#M3653 22C3), CD3 (Biolegend Cat#317336 OKT3), CD19 (Biolegend Cat#302216 HIB19), CD4 (Biolegend Cat#300560 RPA-Ta), CD8a (Biolegend Cat#300906 HIT8a), CD123 (BioLegend Cat#306018 6H6), CD303 (BioLegend Cat#354208 201A), CD45 (BioLegend Cat#304050 HI30), CD11c

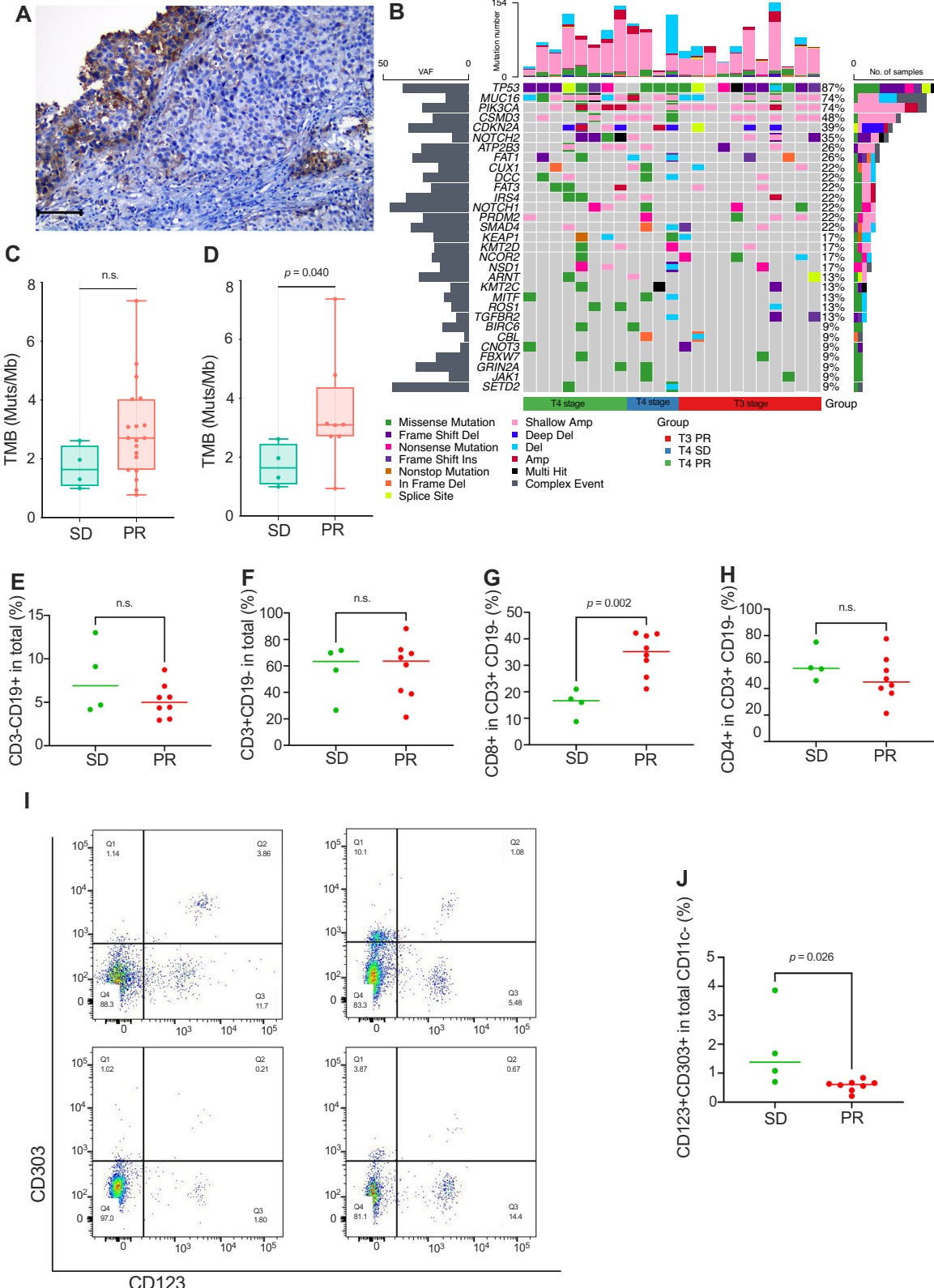

(BioLegend Cat#301608 3.9). Data were acquired using an LSRFortessa (BD Biosciences) and analyzed with FlowJo software (version 10.0).

## Statistical analysis

The sample size was calculated based on the primary endpoint of ORR, which was predicted to be 80% in this study, and the history control value was 60% from our previous data, with 80% detection power and using a two-sided at the formal statistical boundary for the significance of 0.05. Forty-three cases were required to assess the antitumor activity of camrelizumab, plus 15% of cases due to loss to follow-up; overall, 51 participants were required to enroll in this trial. Baseline characteristics, AE, and efficacy followed the intention-to-treat

**Fig. 4 | Tumor mutational burden (TMB), circulating lymphocytes, and DCs correlated with objective response rate (ORR) in patients with T4.**
**A** Representative expression of PD-L1 was tested by immunohistochemistry in primary lesion from biopsy specimens, and the combined positive score (CPS) of this patient was 15% (scale bar = 200 μm). **B** Mutations were analyzed by whole-exome sequencing (WES) of baseline primary tumor lesions from 23 cases. The percentages on the right displayed the proportion of samples harboring a mutation in the genes on the left. The bottom bars displayed ORR from three groups of the T3 stage with PR (*n* = 11 cases), the T4 stage with PR (*n* = 8 cases), and T4 stage with SD (*n* = 4 cases). **C** Comparison of TMB in baseline tumor samples between the patients with PR (*n* = 19 cases) and patients with SD (*n* = 4 cases). **D** Due to four SD patients being at the T4 stage in this subgroup, we deleted the T3 stage population and further analyzed PR and SD patients with the T4 stage. TMB in baseline tumor samples of patients with PR (*n* = 8 cases) was higher than those of patients with SD (*n* = 4 cases). Box plots show distribution of TMB. Lines within box represent the median, the boxes represent quartiles and the whiskers represent the minimum to the maximum. Ns, no statistical significance. Percentage of CD3-CD19+ total B lymphocytes (**E**) and CD3 + CD19- total T lymphocytes (**F**) between patients with SD and PR at the T4 stage. Percentage of CD8+ (**G**) and CD4+ (**H**) T lymphocytes in total CD3 + CD19- T cells between patients with SD and PR at the T4 stage. Percentage CD123 + CD303+ plasmacytoid dendritic cell (pDC) in total CD11c- cells between patients within SD and patients with PR at the T4a stage (**I**), and the corresponding statistical analysis graph was shown (**J**). **E**, **F**, **G**, **H**, **J** the statistical significance was tested using a Mann–Whitney test (two-sided) (SD: *n* = 4 cases; PR: *n* = 8 cases), and the line represent the median. Source data are provided as a Source Data file.

### Table 3 | The relationship between ORR and CPS

| ORR | CPS < 1 | CPS > 1 | *p*\* |
|-----|---------|---------|-------|
| PR | 10 | 8 | 0.257 |
| SD | 3 | 0 | |

Source data are provided as a Source Data file.

\*Two cells have an expected count less than 5, and *p* value tested using Fisher's Exact Test.

principle in this trial. Stata (version 17) was applied to analyze all of the data. A *p*-value less than 0.05 was considered statistically significant.

### Reporting summary

Further information on research design is available in the Nature Portfolio Reporting Summary linked to this article.

## Data availability

The protocol of this study is available as Supplementary Note 1 in the Supplementary Information file. The clinical characteristics, oncologic outcome, adverse events, radiographic, pathological, and exploratory analysis data of this study underlying the results are available in the Supplementary Information and Source data file. The raw sequence data of WES DNA reported in this paper have been deposited in the Genome Sequence Archive (Genomics, Proteomics & Bioinformatics 2021) in National Genomics Data Center (Nucleic Acids Res 2022), China National Center for Bioinformation / Beijing Institute of Genomics, Chinese Academy of Sciences (GSA-Human: HRA006987) [https://ngdc.cncb.ac.cn/gsa-human/browse/HRA006987]. Access to the WES DNA data can be requested for research purpose by completing the application form GSA for Human System and is granted by the Data Access Committee, and the guidance can be found at the website. The de-identified participant data, including radiographic images and blood test results of patients in the current study are available from the corresponding author (L.Z. zhoulent@126.com) for research purposes according to policy of Ethics Committees of Eye & ENT Hospital, Fudan University, after completing the trial for 5 years. The key points of the Committee policy are for scientific human cancer research and are not harmful to people's health. When the access is granted, the data will be available to the requester. The remaining data of this study are available in the Article, Supplementary Information or Source Data file. MOST approval has been obtained for this study. Source data are provided with this paper.

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

## Acknowledgements

We thank all doctors, nurses, and collaborators in this trial who support the clinical diagnosis, treatments, evaluation, and other works, and especially all patients and their families. This work was supported by the National Natural Science Foundation of China (81502343 (H.G.) and 81972529 (L.Z.)), the Clinical Research Plan of SHDC (SHDC2020CR6011 (L.T.) and SHDC2024CRI053 (H.G.)), the Science and Technology Commission of Shanghai Municipality (16411950100 (L.Z.) and 21Y11900100 (H.G.)), and the Shanghai Municipal Key Clinical Specialty (shslczdzk00801 (L.T.)). Hengrui Medicine Co. partially donated the study drug (Camrelizumab, SHR-1210).

## Author contributions

L.Z., S.W., and H.D. conceived and designed the protocol for this study. L.T., L.W., J.W., T.W., Y.S., and C.X. collected the clinical data. L.Z. and S.W. supervised and validated the data and administrated the project. H.G. and S.T. curated and analyzed the data. X.Y. and Y.H. contributed to data acquisition and exploratory analyses. M.Z. and C.W. interpreted the data. H.G. and S.T. wrote the first draft, and L.Z. and S.W. reviewed and edited the manuscript. All authors contributed to reviewing and revising the manuscript and approved the final version. H.G. and S.T. contributed equally to this work.

## Competing interests

The authors declare no competing interests.
