## [Peer Review File · Nature Communications]

Reviewers' Comments:

Reviewer #1:

Remarks to the Author:

Nice phase 2 study, well written. The negative point of this study is that this single-center study is non-randomized.

Reviewer #2:

Remarks to the Author:

1. Why patients got CR or PR received radioimmunotherapy, not chemoradiotherapy?
2. Please tell the causes of death of 9 patients.
3. In the conclusion, only 1-year results were gotten, long term result will need further follow up.
4. In line 78 and 82, what is oncologic outcome means?
5. The English written must be revised.

In 86, previously untreated history and required total laryngectomy, history may delete.
In line 92, Aim to assess the antitumor activity and safety profile of camrelizumab plus TPF of induction chemoimmunotherapy for newly diagnosed LA HSCC patients. The aim of the study was to....

In line 94, the involves may be instead by included.

This trial was performed by the Declaration of Helsinki and the International 106 Conference on Good Clinical Practice guidelines. This will be revised.

In line approved the trial protocol and treatments.

Clinical Target Volume (CTV) 1.6 - 1.9 Gy/fraction. Radiotherapy of cervical lymph node was the same as the primary lesion. Should be the dose of primary...

In line 168 had sufficient tumors? Samples?

induced chemoimmunotherapy should be induction chemoimmunotherapy.

The ORR levels rate should be ORR rate.

Reviewer #3:

Remarks to the Author:

The authors report a single arm phase II trial combining TPF (capecitabine) with camrelizumab, an anti-PD1 agent on 51 patients with LA hypopharyngeal HNSCC. They also present additional biomarker data.

Major comments:

- line 248: the authors should elaborate on reactive cutaneous capillary endothelial proliferation and show a picture. In the discussion should discuss the occurrence of this rare event and make a literature review.
- line 272: CPS is not a percentage but a value. This should be corrected in the manuscript. ORR according to PDL1 expression should be provided, as well as PFS and OS according to PDL1 expression
- line 296: the authors should still mention all negative phase III trials that combined RT and immunotherapy
- the authors should explain why they decided not to give chemoradiation, even in non-responding patients, since this might be a detrimental strategy given all negative results of phase III randomized trials evaluating the addition of immunotherapy to RT
- it would be nice to have the OS and PFS curves according to the treatment strategy adopted following the induction treatment (surgery versus radiotherapy + immunotherapy)

Responses to reviewer's comments

Manuscript NO.: NCOMMS-23-61327A

Reviewer #1 - Biostatistics, clinical trials (Remarks to the Author):

Nice phase 2 study, well written. The negative point of this study is that this single-center study is non-randomized.

Response: Thank you for your comments. In the "Discussion" part we admit the trial's limitation of being a single-center, single-arm, and non-randomized clinical study (page 17, lines 447-453).

Reviewer #2 - Head and neck cancer, clinical trials (Remarks to the Author):

1. Why patients got CR or PR received radioimmunotherapy, not chemoradiotherapy?

Response : Concurrent chemotherapy and radiotherapy is the standard therapy for locally advanced head and neck squamous cell carcinoma (LA HNSCC), and cisplatin-based chemoradiotherapy is demonstrated to be effective when patients get CR or PR after induction therapy. This study was designed in 2019 by our MDT team when neoadjuvant therapy (immune checkpoint inhibitors) was reported to be effective and safe for recurrent or metastatic HNSCC (KEYNOTE-012, 040, and 048). Three trials, including KEYNOTE-412 (NCT03040999), PembroRad (NCT02707588), and JAVELIN Head and Neck 100 (NCT02952586), were designed to evaluate the efficacy and safety of immune therapy plus radiotherapy/chemoradiotherapy for disease of LA HNSCC when we designed our study. Unlike the above three studies, which used radical radiotherapy in combination with drugs, we adopted the strategy of induction therapy screening, and for patients who were effectively treated after induction chemoimmunotherapy, there were three choices of concurrent radiotherapy: 1) platinum-based chemoradiotherapy; 2) platinum-based chemotherapy + immunotherapy; and 3) radioimmunotherapy. Considering the continuity of immunotherapy from induction to maintenance

therapy and the toxic reaction of concurrent radiotherapy, the third choice, radioimmunotherapy, was chosen. The Ethics Committee approved this protocol. Although the subsequent three prospective studies did not confirm the benefits of concurrent immunotherapy and radiotherapy, the short-term efficacy and safety of radioimmunotherapy in our study are promising, and long-term follow-up data will be updated. We improved the discussion in the revised version of the manuscript (page 13, line 359; page 14, line 360-365).

2. Please tell the causes of death of 9 patients.

Response: Thank you for your suggestion, and we added this part based on your suggestion in the revised manuscript. Eight patients died at the time of analyses in this study. We reported the causes of these eight patients in Figure 1 (CONSORT) and results. Five patients died from progression disease, two patients died from uncontrolled disease, and one patients died from local recurrence and regional metastasis (page 9, line 247; page 10, line 248-249).

3. In the conclusion, only 1-year results were gotten, long term result will need further follow up.

Response: We agree that this trial has a few limitations that have to be considered. The long-term oncologic outcomes of this regimen remain uncertain, but this trial is ongoing, and future information will be followed up and updated (page 17, line 447-453).

4. In line 78 and 82, what is oncologic outcome means ?

Response: We explained this problem based on your comment in the revised manuscript. Oncologic outcomes include objective response rate, survival outcome, and laryngeal function preservation (page 3-4, line 79-84).

5. *The English written must revised.*

In 86, previously untreated history and required total laryngectomy, history may delete.

In line 92, Aim to assess the antitumor activity and safety profile of camrelizumab plus TPF of induction chemoimmunotherapy for newly diagnosed LA HSCC patients. The aim of the study was to....

In line 94, the involves may be instead by included.

This trial was performed by the Declaration of Helsinki and the International 106 Conference on Good Clinical Practice guidelines. This will be revised.

In line approved the trial protocol and treatments.

Clinical Target Volume (CTV) 1.6 - 1.9 Gy/fraction. Radiotherapy of cervical lymph node was the same as the primary lesion. Should be the dose of primary...

In line 168 had sufficient tumors? Samples?

induced chemoimmunotherapy should be induction chemoimmunotherapy.

The ORR levels rate should be ORR rate.

Response: Thank you for your comments, and we accepted all of these problems and improved incorrect parts of the manuscript. We used red font to highlight all the revised parts in our manuscript (page 4, line 87, line 93, line 96, page 5, line 109-110, line 129, page 7, line 172; page 9, line 243; page 17, line 455; page 30 and 31).

Reviewer #3 - Head and neck cancer, clinical trials (Remarks to the Author):

The authors report a single arm phase II trial combining TPF (capecitabine) with camrelizumab, an anti-PD1 agent on 51 patients with LA hypopharyngeal HNSCC. They also present additional biomarker data.

Major comments:

- line 248: the authors should elaborate on reactive cutaneous capillary endothelial proliferation and show a picture. In the discussion should discuss the occurrence of this rare event and make a literature review.

Response: We accept your suggestions, and changed this part based on your suggestion in the revised manuscript. The potential pathway of camrelizumab promoting RCCEP is that it activates CD4+ T cells, increases IL-4 levels in T helper 2 cytokines, and stimulates CD163+ M2 macrophage differentiation and capillary endothelial cell proliferation by releasing vascular endothelial growth factor. In this study, the incidence of camrelizumab treatment related RCCEP was 90%, which may be higher than those of the reported data from previous studies, ranging from 58% to 88%. Interesting, the event of RCCEP may generate a promising objective response rate and better survival outcome, and this occurrence may be used as a factor in predicting the efficacy of camrelizumab immunotherapy in clinical treatment (page 15, line 411-415; page 16, line 416-420).

A

B

C

D

Figure 1. RCCEP from patient of No. 27. A. RCCEP with grade 2 occurred after C2 immunotherapy. B. RCCEP sustained after C4 immunotherapy. C and D. RCCEP recovering period (supplementary Fig. 2, page 33, line 681-683).

- line 272: CPS is not a percentage but a value. This should be corrected in the manuscript. ORR according to PDL1 expression should be provided, as well as PFS and OS according to PDL1 expression

Response: We apologize for this problem, and have changed it in the revised manuscript. We provided the analysis table and figure here and the results of statistical analyses in the revised version of manuscript in the “Result” part. However, these analyses results were not statistical significant (page 11, line 281-286).

Table 1. The relationship between ORR and CPS in this study.

ORR	CPS < 1	CPS > 1	p^*
PR	10	8	0.257
SD	3	0	

*Two cells have expected count less than 5, and p value tested from Fisher's Exact Test.

Figure 2. The OS curves of CPS <1 (PDL1_expression = 0) and CPS > 1 (PDL1_expression = 1). Only 21 patients with high quality of immunohistochemistry results were included in this analysis. Log-rank test: $p = 0.3491$. This statistical analysis from Stata software.

Figure 3. The PFS curves of CPS <1 (PDL1_expression = 0) and CPS > 1 (PDL1_expression = 1). Only 21 patients with high quality of immunohistochemistry results were included in this analysis. Log-rank test $p = 0.7588$. This statistical analysis from Stata software.

- line 296: the authors should still mention all negative phase III trials that combined RT and immunotherapy

Response: We discussed this problem based on your suggestion in the revised manuscript. The aim of KEYNOTE-412 (NCT03040999) was to determine the efficacy and safety of pembrolizumab plus cisplatin plus radiation vs. placebo plus CRT in patients with LA HNSCC, and the primary endpoint was event-free survival (EFS). However, the median EFS was not reached (95% CI: 44.7 months - not estimable) in the pembrolizumab group, while in the placebo group it was 46.6 months (95% CI: 27.5 months - not estimable) (HR=0.83). The aim of PembroRad (NCT02707588) was to assess pembrolizumab and radiotherapy vs. cetuximab and radiotherapy in patients with LA HNSCC, and

the primary endpoint was locoregional control (LRC) rate. The 15-month LRC rate was 59% with cetuximab plus RT and 60% with pembrolizumab plus RT. No difference of LRC between concurrent pembrolizumab plus RT and cetuximab plus RT, and no difference in overall survival or progression-free survival between these two treatments at 2-year follow-up. The aim of JAVELIN Head and Neck 100 was to evaluate whether avelumab plus chemoradiotherapy could improve survival outcome in patients with LA HNSCC than chemoradiotherapy alone, and the primary endpoint was progression-free survival (PFS). However, median follow-up for PFS was 14.6 months (IQR 8.5 - 19.6) in the avelumab group and 14.8 months (11.6 - 18.8) in the placebo group. Median PFS was not reached (95% CI 16.9 months - not estimable) in the avelumab group and not reached (23.0 months - not estimable) in the placebo group (stratified hazard ratio 1.21 [95% CI 0.93 - 1.57] (page 12, line 309-315).

- the authors should explain why they decided not to give chemoradiation, even in non-responding patients, since this might be a detrimental strategy given all negative results of phase III randomized trials evaluating the addition of immunotherapy to RT

Response : Concurrent chemotherapy and radiotherapy is the standard therapy for locally advanced head and neck squamous cell carcinoma (LA HNSCC), and cisplatin-based chemoradiation is demonstrated to be effective when patients get CR or PR after induction therapy. This study was designed in 2019 by our MDT team when neoadjuvant therapy (immune checkpoint inhibitors) was reported to be effective and safe for recurrent or metastatic HNSCC (KEYNOTE-012, 040, and 048).

Three trials, including KEYNOTE-412 (NCT03040999), PembroRad (NCT02707588), and JAVELIN Head and Neck 100 (NCT02952586), were designed to evaluate the efficacy and safety of immune therapy plus radiotherapy/chemoradiotherapy for disease of LA HNSCC when we designed

our study. Unlike the above three studies, which used radical radiotherapy in combination with drugs, we adopted the strategy of induction therapy screening, and for patients who were effectively treated after induction chemoimmunotherapy, there were three choices of concurrent radiotherapy: 1) platinum-based chemoradiotherapy; 2) platinum-based chemotherapy + immunotherapy; and 3) radioimmunotherapy. Considering the continuity of immunotherapy from induction to maintenance therapy and the toxic reaction of concurrent radiotherapy, the third choice, radioimmunotherapy, was chosen. For non-responding patients to induction chemoimmunotherapy, recognized as non-sensitive to immunotherapy, then underwent surgery and radiotherapy or chemoradiotherapy if they had high-risk pathological factors. Our MDT team widely discussed this protocol and applied it to the Ethics Committee. Although the subsequent three prospective studies did not confirm the benefits of concurrent immunotherapy, the short-term efficacy and safety of concurrent immunotherapy in this study are promising, and long-term follow-up data will be updated (page 13, line 359; page 14, line 360-365).

- it would be nice to have the OS and PFS curves according to the treatment strategy adopted following the induction treatment (surgery versus radiotherapy + immunotherapy)

Response: Thanks for the comments. We analyzed the OS and PFS curves according to the treatment strategy adopted following the induction chemoimmunotherapy and found no difference in OS and PFS between surgery and radioimmunotherapy. We provided the figure here and the statistical analysis results in the manuscript's revised version in the "Result" section (page 10, line 271-274).

Figure 4. The OS curves of surgery versus radiotherapy + immunotherapy. ORR = 1: surgery + postoperative chemoradiotherapy, ORR = 0: radiotherapy + immunotherapy. Log - rank test $p = 0.0552$. This statistical analysis from Stata software.

Figure 5. The PFS curves of surgery versus radiotherapy + immunotherapy. ORR = 1: surgery + postoperative chemoradiotherapy, ORR = 0: radiotherapy + immunotherapy. Log - rank test $p = 0.1851$. This statistical analysis from Stata software.

Reviewers' Comments:

Reviewer #2:

Remarks to the Author:

The authors have revised the manuscript and answered all questions.

Reviewer #3:

Remarks to the Author:

The ORR par CPS and management (surgery versus not) should be provided in the text (not only the p values).

In the text, the median PFS and OS for each of these subgroups should be provided (not only the p values). The PFS and OS curves should be in a supplementary document.

Although not statistically significant, the surgical approach seems more efficient. This should be stated and also discussed in the discussion.

Responses to reviewers' comments
Manuscript NO.: NCOMMS-23-61327B

Reviewer #2 (Remarks to the Author):

The authors have revised the manuscript and answered all questions.

Response: Thank you for your comments and suggestions, and these are very helpful and meaningful for our manuscript.

Reviewer #3 (Remarks to the Author):

The ORR par CPS and management (surgery versus not) should be provided in the text (not only the p values).

In the text, the median PFS and OS for each of these subgroups should be provided (not only the p values). The PFS and OS curves should be in a supplementary document.

Although not statistically significant, the surgical approach seems more efficient. This should be stated and also discussed in the discussion.

Response: We accepted your suggestions, and changed them based on your suggestion in the revised manuscript. The relationship between ORR and CPS was analyzed and showed in the Table 3, and survival curves were showed in Supplementary Fig. 3. The OS and PFS rates of management (surgery versus radioimmunotherapy) after induction therapy was showed in the result part, and survival curves were displayed in Supplementary Fig. 3.

Fifty-one patients enrolled and nine patients died at the time of analysis in this study, the medium OS and PFS cannot be calculated currently and we will update this data in future. Here, we displayed the 1-year and 2-year OS and PFS of subgroups in the manuscript. After induction chemoimmunotherapy, PR patients underwent radioimmunotherapy, and SD patients underwent surgery plus adjuvant therapy. The estimated 1-year OS and PFS rates in patients with PR were 86.9% (95% CI, 68.7% to 94.9%) and 79.8% (95% CI, 61.9% to 89.9%). The estimated 2-year OS and PFS rates in patients with PR were 80.0% (95% CI, 55.9% to 91.7%) and 72.8% (95% CI, 50.1% to 86.5%). Only nine patients showed SD, and the estimated 1-year and 2 year OS and PFS rates in these patients were all 64.7% (95% CI, 25.6% to 87.0%). The OS and PFS of patients undergoing radioimmunotherapy may be better than patients undergoing surgery plus adjuvant therapy after induction therapy, but no statistical difference was detected. We also stated and discussed this problem in the discussion part.

We used red font to highlight all the revised parts in our manuscript